# The Ability of Different Tea Tree Germplasm Resources in South China to Aggregate Rhizosphere Soil Characteristic Fungi Affects Tea Quality

**DOI:** 10.3390/plants13152029

**Published:** 2024-07-24

**Authors:** Xiaoli Jia, Shaoxiong Lin, Qi Zhang, Yuhua Wang, Lei Hong, Mingzhe Li, Shuqi Zhang, Tingting Wang, Miao Jia, Yangxin Luo, Jianghua Ye, Haibin Wang

**Affiliations:** 1College of Tea and Food, Wuyi University, Wuyishan 354300, China; jiaxl2010@126.com (X.J.);; 2College of Life Science, Longyan University, Longyan 364012, China; 3College of JunCao Science and Ecology, Fujian Agriculture and Forestry University, Fuzhou 350002, China

**Keywords:** *Camellia sinensis*, ecological memory, deep machine learning, nutrient transformation, quality

## Abstract

It is generally recognized that the quality differences in plant germplasm resources are genetically determined, and that only a good “pedigree” can have good quality. Ecological memory of plants and rhizosphere soil fungi provides a new perspective to understand this phenomenon. Here, we selected 45 tea tree germplasm resources and analyzed the rhizosphere soil fungi, nutrient content and tea quality. We found that the ecological memory of tea trees for soil fungi led to the recruitment and aggregation of dominant fungal populations that were similar across tea tree varieties, differing only in the number of fungi. We performed continuous simulation and validation to identify four characteristic fungal genera that determined the quality differences. Further analysis showed that the greater the recruitment and aggregation of *Saitozyma* and *Archaeorhizomyces* by tea trees, the greater the rejection of *Chaetomium* and *Trechispora*, the higher the available nutrient content in the soil and the better the tea quality. In summary, our study presents a new perspective, showing that ecological memory between tea trees and rhizosphere soil fungi leads to differences in plants’ ability to recruit and aggregate characteristic fungi, which is one of the most important determinants of tea quality. The artificial inoculation of rhizosphere fungi may reconstruct the ecological memory of tea trees and substantially improve their quality.

## 1. Introduction

Memory is the storage of information about past experiences by organisms, and organisms perform memory behaviors every day. In contrast to higher animals, which store memories in specialized brain cells, plants lack an organized brain or nervous system. Research studies have consistently pointed out that only animals are capable of complex memory behaviors. The proposition that plants may possess memory behaviors initially created a paradigm shift in perception and has since been the subject of significant debate among scholars [1]. Plants can “perceive and respond to their environment” either through inherited knowledge from their parents or through the use of past experiences, allowing them to adapt more effectively to their surroundings [2,3]. Despite the limitations imposed by their immobility, plants have been observed to employ memory behaviors, emitting specific chemical signals that attract mobile organisms to meet their needs [4,5,6]. Plants also seek cues through memory to respond effectively to danger. Karban [7] and Spoel and Dong [8] have demonstrated how plants use memory to initiate familiar pathways to protect themselves from harm. For example, after a plant has overcome its first instance of drought stress, the strategies that it adopts to combat stress become a memory, and, when faced with a subsequent drought, the plant can deal with it more effectively and with a greater likelihood of survival [9]. In summary, plants possess a form of memory that they use to better acclimate to their surroundings and optimize their growth patterns [10]. Therefore, it is important to explore how plants, through memory, adapt to changing living environments and thus safeguard their growth.

Soil plays a crucial role in plant growth because it is a highly complex system that includes water, nutrients, animals and microorganisms that interact with the plant and work together to support plant growth [11]. Microorganisms comprise the overwhelming majority of soil life, regulating a wide range of biological reactions and driving the continuous biogeochemical cycles that control soil material transformations and ultimately regulate plant growth [12,13]. Soil fungal communities play an important role in plant health and productivity, and plant–fungi interactions promote fungi evolution and altered function [14]. The coevolution and complex interactions between plants and fungi over a long period have led to their formation of unique memories of each other [15]. To better adapt to their environments, plants and fungi have developed a mutual drive for memory behavior, and this behavior is a result of their shared understanding and adaptability to their surroundings [16]. Moreover, this memorized behavior is referred to as “microbial-induced soil genetics”, where the microbial community within plant roots develops from the moment of seed germination and is transferred from generation to generation [17].

At the moment that a seed germinates, the root system begins recruiting and aggregating its remembered microbes, reorganizing the dispersed microbial community into a remembered dominant microbial assemblage to safeguard its growth in response to current and possible future changes [18]. Aggregated microbial functions are reorganized, strengthening the plant’s defenses against changing environments and enabling more effective support for plant growth [19]. It is clear that there is a close association of the specific ecological memory between plants and soil fungi and plant growth. However, the effects of this memory behavior on plant growth and fungal communities are often overlooked when studying plant–microbe community interactions. Ecological memory may be able to more accurately explain how plants change the structure of fungal communities and how fungal communities in turn influence plant growth [20].

Tea tree (*Camellia sinensis*) is a plant in the Camelliaceae family, which has a high economic value and is important for promoting agricultural development. In China, tea tree germplasm resource nurseries have been established in all major tea-producing provinces, dedicated to the integration and conservation of different tea tree varieties to carry out in-depth research and development, so as to cultivate excellent tea tree varieties and promote the sustainable development of the tea industry [21,22]. It is well known that different tea tree varieties have significant differences in growth and quality, especially in terms of quality performance [23,24,25,26]. As an important economic crop in China, a large number of Chinese scholars have conducted in-depth studies on the causes of quality differences in tea tree varieties from multiple perspectives, such as genome, transcriptome, metabolome, genetics, etc., and it is generally recognized that the differences are mainly caused by differences in genetic physiological functions [27,28,29,30,31]. From the point of view of the internal mechanism of the tea tree itself, genetic differences lead to differences in physiological mechanisms, which in turn affect the quality of tea leaves. This means that if the tea tree does not have a good genetic background, regardless of how it is managed and regulated in the later stages, it will be difficult to obtain excellent quality. However, the reality is that the differences in the quality of different varieties of tea trees are the result of interactions between the genotype of the tea tree and the environment, especially the interaction of the tea tree with soil microorganisms. When plants are planted, rhizosphere secretions selectively drive the evolution of the microbial community structure, thus maintaining the relative stability of the soil in terms of community structure, diversity, and function [32]. This creates memory behavior between microbes and plants, which alters nutrient cycling in the soil and thus affects plant quality [33,34,35,36,37]. We propose the hypothesis that there is mutual mnemonic behavior between tea trees and rhizosphere soil fungi and that this behavior leads to the possibility that different tea tree germplasm resources may be similar in dominant fungal populations that they recruit and aggregate, despite their genetic differences. In contrast, different varieties of tea trees may differ in their ability to recruit and aggregate fungi, which will have an impact on soil nutrient cycling and thus tea quality.

Based on the above hypotheses, rhizosphere soil and leaf samples were collected from 45 tea tree germplasm resources (Appendix A) from the Tea Tree Germplasm Resource Nursery of Wuyi University, Wuyishan City, Fujian Province, China. Soil fungal diversity was analyzed using high-throughput sequencing technology, and dominant population distribution was also explored. We also performed an unsupervised K-means clustering analysis based on fungal abundance, seeking to classify tea germplasm resources from a fungal perspective. After the classification was completed, we applied the OPLS-DA model combined with K-nearest neighbor (KNN), support vector machine (SVM), back-propagation neural network (BPNN), random forest (RF), extreme gradient boosting (XGboost) and other machine and deep learning techniques to verify the accuracy of the classification. Finally, we identified the characteristic fungi to distinguish 45 tea tree germplasm resources and verified them by *q*RT-PCR. On this basis, this study first explored the functions of the characteristic fungi and analyzed their relationship with soil nutrients and tea quality. Overall, this study provides a new perspective on the quality differences among tea tree germplasm resources from the perspective of ecological memory. This study has important implications for future cultivation practices aimed at regulating the quality of different varieties of tea trees or other plants through microorganisms.

## 2. Results and Discussion

### 2.1. Basic Information on the Rhizosphere Soil Fungal Community of Tea Tree

We collected rhizosphere soil from 45 tea tree germplasm resources (Appendix A) for amplicon sequencing analysis of fungal communities (https://www.ncbi.nlm.nih.gov/sra/PRJNA1045082, accessed on 26 November 2023) and obtained 5,430,075 Mb of clean tags (Appendix A). Sequence lengths were mainly distributed from 0 to 540 bp, especially from 0 to 260 bp, and the proportion of high-quality sequences reached 80.19% (Appendix A). After the clean tags’ cluster analysis, 28592 OTUs were obtained, and the number of OTUs of rhizosphere soil fungi from 45 tea tree germplasm resources was distributed between 403 and 818 (Appendix A), of which 55 OTUs were similar (Appendix A). The rarefaction (Appendix A), Shannon–Wiener coefficients (Appendix A), rank abundance (Appendix A) and species accumulation curves (Appendix A) of the OTUs indicated that the amount of sequencing data was sufficient to reflect the rhizosphere soil fungal community of tea trees, and even if the number of samples increased further, the number of new fungal species detected was very small. Simpson coefficients (distribution range 0.56–0.97, mean 0.92) and Shannon coefficients (distribution range 2.68–6.62, mean 5.45) of the OTUs showed that soil was richer in fungal diversity (Appendix A). Chao1 (distribution range 544.26–1316.85, mean 995.62) and PD whole tree indexes (distribution range 72.07–140.68, mean 115.03) showed that fungal community abundance varied significantly among samples (Appendix A).

### 2.2. Tea Germplasm Resources Recruited and Aggregated Similar Dominant Fungal Populations

Soil is extremely rich in microbial populations, which have a significant impact on plant growth and development [38]. Initially, soil microorganisms lived normally in their respective domains, and the presence of plants caused changes in the soil microbial community structure, resulting in a clustering effect, especially in the plant rhizosphere [39,40,41]. This phenomenon reflects the selectivity of plant root secretions within fungal communities, leading to characteristic microbial clusters, and is the result of interactions between plants and microorganisms and their coevolution [42,43]. Thus, the closer to the root system, the more pronounced the clustering effect of the characteristic microorganisms, and the rhizosphere is the best area to study plant-microbe interactions [44]. Based on OTU species annotation and classification results, we found that a total of 549 genera of fungi were present in rhizosphere soils of 45 tea tree germplasm resources (BioProject ID: PRJNA1045082), with extremely high diversity abundance. Among them, there were 20 genera with relative abundance of 1% or more, accounting for total fungal abundance ranging from 65.636% to 98.739%, with a mean of 92.086% (Figure 1). Excluding those that were unidentified, 19 genera accounted for total fungal abundance ranging from 36.663% to 91.309%, with a mean of 64.238% (Figure 1). Bickford et al. [45] studying the rhizosphere microbial communities of different species of *Phragmites australis*, concluded that differences in microbial communities were not related to plant lineage but focused on differences in the ability of different species to cluster characteristic microorganisms. Similar results were found in *Gossypium* sp. [46] and *Glycyrrhiza uralensis* [47], i.e., rhizosphere-soil-dominant microbial populations were similar among different varieties of the same plant, but there was a significant difference in abundance, which led to differences in soil nutrient cycling capacity, which in turn affected the plant quality. The results of this study indicated that different tea tree varieties recruited and aggregated similar populations of dominant fungi, while there were differences in the number of fungi, i.e., differences in the ability of different tea tree varieties to recruit and aggregate fungi.

### 2.3. Screening and Validation of Characteristic Fungi

Since there are differences in the ability of different varieties of tea trees to recruit and aggregate fungi, what exactly are these fungi? To address this query, we used unsupervised K-means clustering for automatic classification and discrimination based on the abundance of the rhizosphere soil fungi in the 45 tea tree germplasm resources. We found that 45 tea tree germplasm resources could be categorized into three groups, which we defined as group A, group B, and group C (Figure 2A), of which 11 tea tree varieties were classified in group A, 25 in group B, and nine in group C (Appendix A). Initially, groups A, B, and C were subjected to Bray-Curtis heat map analysis, and it was clear that there were some differences between the three groups (Appendix A). Therefore, we further used supervised OPLS-DA to construct a model of groups A, B, and C to screen for fungi with key differences between the three groups (Figure 2B), and a total of 181 genera of key fungi were obtained (Appendix A). The Bray-Curtis heatmap analysis of key fungi showed more pronounced differentiation between groups A, B, and C (Appendix A).

We used five machine and deep learning algorithms (KNN, BPNN, SVM, RF, and XGboost) to validate the accuracy of 181 genera of key fungi by distinguishing between groups A, B, and C and obtaining feature importance values for different fungi. The analysis revealed that the average accuracy of the five machine and deep learning simulated classifications reached 91.87% (confusion matrix), and the classification accuracy for group A, group B, and group C reached 0.90, 0.90, and 0.92 (ROC curves), respectively (Figure 3A, Appendix A). The top 30 fungal genera ranked by their feature importance values were derived from RF and XGboost (Appendix A), and 25 fungal genera were found to be shared (Figure 3B). A machine and deep learning simulation with 25 genera of characteristic fungi was performed again to verify the classification accuracy, which was found to be 89.60% on average, and the classification accuracy for group A, group B, and group C reached 0.89, 0.88, and 0.94, respectively (Figure 3C, Appendix A). The abundance analysis of the characteristic fungi revealed (Figure 3D, Appendix A) that, in terms of the abundance of 25 genera of fungi in group A, group B, and group C, 22 genera showed a significant difference (*p* < 0.05), and three genera showed no significant difference. Secondly, we found that, among the 22 fungal genera with significant differences, the largest abundance was found in *Saitozyma* (range 1.91% to 66.06%, mean 17.20%), *Archaeorhizomyces* (range 0.04% to 21.14%, mean 3.06%), *Trechispora* (range 0% to 65.88%, mean value 5.46%) and *Chaetomium* (range 0.08% to 14.20%, mean value 3.10%) (Appendix A).

Based on the 22 characteristic fungal genera with significant differences, unknown and temporarily unsearchable fungi in the NCBI database were removed. We used *q*RT-PCR to quantify 20 fungal genera and found significant differences in the numbers of these genera in group A, group B, and group C (Figure 4A, Appendix A). This result was consistent with the abundance analysis described in the section above (Figure 3C, Appendix A). The data obtained by *q*RT-PCR were again validated by machine and deep learning simulation, the average accuracy of the five machine and deep learning simulated classifications was found to be 98.80%, and the classification accuracy for group A, group B, and group C was 1.00, 0.98, and 0.97, respectively (Figure 4B, Appendix A). Moreover, we found that, after the quantification of the fungi, *Saitozyma* (87.53~438.91 × 10^3^ cell/g·soil, mean 148.88 × 10^3^ cell/g·soil), *Archaeorhizomyces* (45.61~206.82 × 10^3^ cell/g·soil, mean 68.79 × 10^3^ cell/g·soil), *Trechispora* (25.22~478.14 × 10^3^ cell/g·soil, mean 65.57 × 10^3^ cell/g·soil), and *Chaetomium* (4.43~53.18 × 10^3^ cell/g·soil, mean 19.39 × 10^3^ cell/g·soil) were the most abundant, and the numbers of *Archaeorhizomyces* and *Saitozyma* followed the order of group C > group B > group A, while the numbers of *Chaetomium* and *Trechispora* followed the order of group A > group B > group C (Figure 4C, Appendix A). This result was consistent with the results of the above abundance analysis (Figure 3E). These results explained the doubts raised and showed that, when analyzed from the fungal point of view, 45 tea tree germplasm resources could be classified into three groups. In terms of distinguishing them, four fungal genera were found to be critical, all of which were dominant populations, and they differed significantly in abundance. It has been reported that plants often induce and aggregate characteristic microorganisms during growth by releasing root secretions, which in turn change soil physicochemical indexes and affect plant growth [48,49]. It can be seen that different tea tree germplasm resources may exhibit differences in the number and type of their root secretions, and this leads to differences in the number of aggregated *Archaeorhizomyces*, *Saitozyma*, *Chaetomium*, and *Trechispora*, which in turn affects the growth of tea trees.

Based on the existing literature, *Archaeorhizomyces* [50,51,52], *Saitozyma* [53,54,55], *Chaetomium* [56,57,58], and *Trechispora* [59,60,61] all play an important role in the nutrient cycling of rhizosphere soil by altering the content of available nutrients in the soil and affecting plant quality. Can it also be hypothesized that there are significant differences in the rhizosphere soil’s available nutrients among the 45 tea tree germplasm resources? This issue needs to be further explored.

### 2.4. Soil Available Nutrient Content

Based on the above, we attempted to determine the available nutrient content (available nitrogen, available phosphorus, and available potassium) of the rhizosphere soils of 5 tea tree germplasm resources (Appendix A) and analyzed them using unsupervised K-means clustering and machine learning. Surprisingly, unsupervised K-means clustering also classified the 45 tea germplasm resources into three groups (defined as group A1, group B1, and group C1) (Figure 5A, Appendix A), which overlapped with the aforementioned group A, group B, and group C (Appendix A). Five machine learning simulations were used to verify classification accuracy, and it was found that the average accuracy reached 99.07%, with group A1, group B1, and group C1 reaching 0.99, 0.99, and 1.00, respectively (Figure 5B, Appendix A). More surprisingly, the differential analysis of available nutrient content in group A1, group B1, and group C1 showed that available nitrogen, available phosphorus, and available potassium in group C1 were significantly higher than those in group B1, while those in group B1 were significantly higher than those in group A1 (Figure 5C, Appendix A). Combined with the above findings, this result suggested that the key differences between the three groups into which 45 tea tree germplasm resources were divided were significant differences in recruitment and aggregation capacity of four characteristic fungal genera, namely *Archaeorhizomyces*, *Saitozyma*, *Chaetomium*, and *Trechispora*. Moreover, such differences resulted in significant differences in available nutrient content in tea tree rhizosphere soils. The content of available nutrients in the soil is significantly and positively correlated with tea quality, and increasing the content of available nutrients in the soil could improve tea quality [62,63,64,65]. The improvement of tea quality is of great significance in enhancing the economic benefits of tea, and tea polyphenols, theanine, and caffeine are important indexes for the evaluation of tea quality. Numerous scholars use the content of these three indexes to evaluate tea quality; high content indicates good quality, while low content indicates poor quality [66,67,68]. In view of this, we hypothesized that the 45 tea tree germplasm resources may also differ significantly in quality.

### 2.5. Tea Quality Index Content Analysis

Based on the above hypothesis, we tried again to determine the content of quality indexes (tea polyphenols, caffeine, and tea polyphenols) in leaves from 45 tea germplasm resources (Appendix A), which were again analyzed using unsupervised K-means clustering and machine learning. Our hypothesis was confirmed. Unsupervised K-means clustering also classified the 45 tea germplasm resources into three groups (defined as group A2, group B2, and group C2) (Figure 6A, Appendix A), and these three groups were fully consistent with group A, group B, and group C (Appendix A) and group A1, group B1, and group C1 (Appendix A). Five machine and deep learning methods were used to simulate and validate the classification accuracy, and it was found that the average accuracy reached 99.93%, with that of group A2, group B2, and group C2 reaching 1.00, 1.00, and 1.00, respectively (Figure 6B, Appendix A). More surprisingly, the differential analysis of leaf quality index content between group A2, group B2, and group C2 showed that tea polyphenols, theanine, and caffeine levels in group C2 were significantly higher than those in group B2, while those in group B2 were significantly higher than those in group A2 (Figure 6C, Appendix A). Combined with the above findings, this result suggested that the three groups in 45 tea tree germplasm resources were clearly divided, and their ability to recruit and aggregate the four characteristic fungal genera determined the available nutrient content of the rhizosphere soils of the tea tree and tea quality.

### 2.6. Correlation Analysis among Soil Characteristic Fungi, Available Nutrients, and Tea Quality Indexes

Based on the above, we correlated quantitative data from the four characteristic fungal genera with available nutrient content and quality index content of the soil (Appendix A). We found that the fungi associated with group A were mainly *Chaetomium* and *Trechispora*, while those associated with group C were mainly *Archaeorhizomyces* and *Saitozyma*, and group B was located between group A and group C. The available nutrient content of the soil and the content of the tea quality indexes were both related to *Archaeorhizomyces* and *Saitozyma*. Interaction network analysis also showed (Appendix A) that tea quality and soil nutrients were significantly and positively correlated with *Archaeorhizomyces* and *Saitozyma* and significantly and negatively correlated with *Chaetomium* and *Trechispora*. Overall fitting analysis using partial least squares structural equation modeling (PLS-SEM) (Figure 7) revealed that the characteristic fungal genera had an extremely significant and positive effect (*p* < 0.001) on soil available nutrient content, while soil available nutrient content had an extremely significant and positive effect (*p* < 0.001) on the tea quality index content. This result shows that if the root system of the tea tree is biased to recruit aggregates of *Saitozyma* and *Archaeorhizomyces* and excludes *Chaetomium* and *Trechispora*, this positively affects the content of available nutrients in the soil, which in turn improves tea quality.

## 3. Materials and Methods

### 3.1. Sample Collection

All 45 of the different tea tree germplasm resources featured in this study were acquired from the Tea Tree Germplasm Resource Nursery of Wuyi University, Wuyishan City, Fujian Province, China (117°59′51.88″ E, 27°44′16.17″ N). 45 tea tree germplasm resources were planted in the same plot. The tea tree germplasm resource nursery was situated 216 m above sea level, the soil was an acidic red loam, the average annual temperature was 19 °C, the annual rainfall was 1284 mm, the average humidity was 74%, and the age of the planted tea trees was 10 years. In October 2022, a total of 45 tea germplasm resources (Figure 8, Appendix A) were sampled for rhizosphere soil and leaf samples. Soil samples were used to determine available nutrients (available nitrogen, available phosphorus, and available potassium) and to perform the amplification sequencing of fungal diversity and *q*RT-PCR analysis of characteristic fungi. Leaf samples were used to determine tea quality indexes (tea polyphenols, theanine, caffeine). The sampling method for rhizosphere soil of tea trees was the S-type sampling method. Specifically, for each variety of tea trees, eight tea trees were randomly selected, litter on the soil surface was removed, tea trees were gently dug out, the tea tree root system was removed, soil adhering to the root surface was shaken off, and soil still adhering to the roots was collected as tea tree rhizosphere soil, which was replicated after sufficient mixing [69]. While collecting the rhizosphere soil of the tea tree, one bud and two leaves of the tree were collected and thoroughly mixed to create a replicate [68]. Three independent replicates were taken for each tea tree variety.

### 3.2. Fungal ITS Amplicon Sequencing

The E.Z.N.A. Soil DNA Kit (Omega Bio-tek Inc., Norcross, GA, USA) was used to extract genomic DNA from the soil microorganisms in the rhizosphere soil of the tea trees. The DNA was purified using a gel recovery kit from TianGen Biotech Co., Ltd. (Beijing, China) and detected by 1% agarose gel electrophoresis. The DNA quality and concentration were assayed using a Nanodrop 2000 (ThermoFisher Scientific, Inc., Waltham, MA, USA) for PCR amplification.

The amplification of the ITS rDNA of soil fungi was performed by the following method [70]. PCR amplification primers were ITS1 (CTTGGTCATTTAGAGGAAGTAA) and ITS2 (TGCGTTCTTCATCGATGC). The PCR reaction system (total system was 25 μL) included 2 μL of DNA (30 ng), 1 μL of forward primer (5 μM), 1 μL of reverse primer (5 μM), 3 μL of BSA (2 ng/μL), 12.5 μL of 2×Taq Plus Master Mix, and 5.5 μL of dd H_2_O. The PCR reaction program consisted of pre-denaturation at 95 °C for 5 min, denaturation at 95 °C for 45 s, annealing at 55 °C for 50 s, extension at 72 °C for 45 s, 34 cycles, and a final extension at 72 °C for 10 min. PCR products were purified using the Agencourt AMPure XP (Beckman Coulter Life Sciences, Indianapolis, IN, USA) nucleic acid purification kit. The purified PCR products were used for library construction using NEB Next Ultra II DNA Library Prep Kit (New England Biolabs, Ipswich, MA, USA). After the library construction was completed, the library fragment size was detected using Agilent 2100 Bioanalyzer (Agilent Technologies Inc., Santa Clara, CA, USA), and the library concentration was accurately quantified using the ABI StepOnePlus Real-Time PCR System (Applied Biosystems–Thermo Fisher Scientific, Waltham, MA, USA). The libraries were sequenced by Beijing Allwegene Technology Co., Ltd. (Beijing, China) using Illumina Miseq/Nextseq 2000/Novaseq 6000 (Illumina, San Diego, CA, USA) platforms with a sequencing strategy of PE250/PE300.

### 3.3. Bioinformatics Analysis

The amplicons of the fungal ITS rDNA gene sequences were analyzed using Illumina Analysis Pipeline (v2.6). First, Fastq data was filtered and spliced using Trimmomatic software (v0.36) and Pear software (v0.9.6) to remove sequences containing fuzzy base N or quality values below 20; the minimum overlap was set to 10 bp for splicing and the *p*-value to 0.0001 [71]. After splicing, sequences less than 120 bp in length were removed using the Vsearch software (v2.7.1) and chimeric sequences were removed by comparison using the uchime method according to the Unite database (v8.2) to obtain clean tags [72]. The OTU clustering of high-quality sequences was performed using the uparse algorithm of the Vsearch software (v2.7.1), with a sequence similarity threshold of 97% [73]. The OTU sequences were compared with the Unite database (v8.2) using the BLAST algorithm, with the e-value threshold set to le-5, to obtain taxonomic information about the species corresponding to each OTU [74,75].

Based on OTUs and their abundance results, rarefaction, Shannon–Wiener, rank abundance, and species accumulation curves were analyzed. Sequential random sampling was used to construct a rarefaction curve with the number of sampled sequences versus the number of OTUs they could represent. A Shannon–Wiener curve was constructed using the fungal diversity index of each sample at different sequencing depths as a reflection of the fungal diversity of each sample at different sequencing numbers. The OTUs were ranked in ascending order of abundance, and then the rank abundance curve was constructed using the OTU rank as the horizontal coordinate and the number of sequences contained in each OTU as the vertical coordinate. Species accumulation curves were constructed using the sample size as the horizontal coordinate and the rate of emergence of new OTUs under continuous sampling as the vertical coordinate. The α diversity index analysis (Shannon coefficients, Simpson coefficients, Chao1 index, and PD whole tree index) was conducted using the QIIME1 (v1.8.0) software and the results plotted using the Rstudio software (R version 4.2.3).

### 3.4. Construction and Evaluation of Machine and Deep Learning Models

Machine and deep learning are uniquely useful for filtering and classifying big data for validation and have been widely used for classification evaluation. There is no single algorithm for all cases. Therefore, based on the obtained soil fungal abundance, we used five machine and deep learning algorithms, namely K-nearest neighbors (KNN), support vector machine (SVM), back-propagation neural network (BPNN), random forest (RF), and eXtreme gradient boosting (XGboost), to model the 45 tea tree germplasm resources and verify the classification accuracy. All models were constructed using Rstudio software (R version 4.2.3), and evaluation parameter tables were produced in Excel. During the model construction process, this study used stratified sampling technique, i.e., data samples were divided into training sets and test sets at a ratio of 80% and 20%. To ensure the robustness of the models, all models in this study were trained with 100 random iterations, and the training and test sets were randomly combined proportionally in each training iteration. Finally, the model integration analysis was carried out with the results of 100 iterations of training, which integrated all models from the training process into a comprehensive result, improving the stability and accuracy of the model predictions.

In predicting new values, the KNN classification algorithm determines attributes based on the categories of the nearest k points, which are simple, fast, and insensitive to outliers. Selecting the k value will have an impact on the algorithm’s results. When the value of k is small, the overall complexity of the model decreases and it is easy to overfit. When k is large, it selects training set instances that are far away from validation set samples, thus affecting prediction and generating errors [76]. In this study, a tuning grid containing k values, where k values ranged from 1 to 45, was established for cross-validation.

The core idea of the SVM classification algorithm is to find an optimal hyperplane that maximizes the spacing between different classes. By introducing kernel functions, SVM is able to handle nonlinear classification problems [77]. Radial basis kernel function (RBF)-based SVM was used in this study. In order to obtain the optimal model parameters, a parameter grid containing the values of C and σ was defined, where both the C and σ values ranged from 10^−3^ to 10^3^, with an increase of one order of magnitude at a time, and cross-validation was performed.

The BPNN classification algorithm performs classification prediction through forward and backward information propagation, i.e., information is propagated forward through layer-to-layer connections, and the back-propagation algorithm is used for error back-propagation and weight updating, to minimize the error between the predicted output and the actual output [78]. In this study, a back-propagation neural network model with a hidden layer was designed and constructed. The model had a hidden layer between the input and output layers to capture and process complex features in the data, serving to enhance the model’s learning and representation capabilities.

RF is a comprehensive supervised classification algorithm that can effectively solve the large error and overfitting problems that can occur in decision trees [79]. The model consists of many decision trees that are unrelated to each other. After obtaining the forest, the model judges or predicts new samples by judging each decision tree in the forest separately to distinguish which category the sample belongs to and determine which category has the highest number of selections, thus providing a judgment on the sample category. Deciding how many trees should be included in this model is crucial [80]. In this study, the initial number of trees was set to 181, and a parameter space containing mtry (the number of variables considered per split) was also defined, with mtry taking values from 1 to 181. We explored variations in model complexity by dividing this parameter space into subspaces and cross-validating it across subspaces. In addition, this study used the RF model to derive the importance scores of each variable for model construction in order to evaluate the magnitude of each variable’s contribution to model construction and then filter out the most important characteristic variables.

XGboost is a widely used gradient enhancement algorithm that is efficient, flexible, and portable; it helps solve classification and regression tasks. XGboost uses an additive strategy to incrementally add trees and optimizes the objective function at each step using the gradient boosting principle, where the model tries to better correct residuals of all previous trees at each step, resulting in more accurate predictions [81]. To determine the optimal model parameters, this study defined a parameter grid that included nrounds (the number of augmentation iterations), max_depth (the maximum depth of the tree), eta (the learning rate), colsample_bytree (the ratio of the subsamples of the features used in the construction of each tree), min_child_weight (the minimum weight of the child nodes), and subsample (the ratio of the subsamples of the training data) to be used for cross-validation and parameter optimization. In addition, this study derived the importance scores of the impacts of all variables on model construction through the model constructed by XGboost in order to evaluate the value of the contribution of each variable to model construction and thus obtain the most important characteristic variables.

All models created underwent a preliminary evaluation of their performance, which was carried out using the confusion matrix and ROC curves. Confusion matrices are a popular tool for evaluating the classification performance of each group by analyzing the correspondence between the true category of sample data and the predicted results. They provide a straightforward, intuitive, and quantifiable approach to assessing classification specificity [82]. In this study, we used the model integration technique to compute the total confusion matrix of the whole set of models. This technique allows us to combine predictions made by individual models and produce a more robust and precise combined prediction [83]. This method also allows the calculation of the confusion matrix for the model as a whole, thus providing an overall assessment of the predictive performance of all models. This research further explored the model’s performance by constructing ROC curves for each machine learning algorithm after each iteration. ROC curves are a frequently used tool for evaluating the effectiveness of binary classifiers, providing a comprehensive depiction of the model’s performance by demonstrating the relationship between true positive and false positive rates across different classification thresholds. The model’s generalizability was also evaluated using the area under the curve (AUC) in the ROC curve. AUC values can range from 0 to 1.0, and a higher AUC is generally correlated with improved model quality [84,85].

### 3.5. qRT-PCR Analysis of Soil Characteristic Fungi

Based on previous studies and analyses, a total of 22 genera of characteristic fungi with significant differences were obtained in this study. After removing unknown and temporarily unsearchable fungi from the NCBI database, this study further quantified the number of fungi in 20 genera, with three independent replicates set up for each sample. Soil microbial genomes were extracted using the E.Z.N.A. Soil DNA Kit instructions (Omega Bio-tek, Inc., Norcross, GA, USA). DNA was purified and detected by 1% agarose gel electrophoresis using a gel recovery kit from TianGen Biotech Co. Purified DNA was used for *q*RT-PCR analysis of fungi. The primers used for *q*RT-PCR analysis of 20 genera of fungi are shown in Appendix A. The PCR reaction programs were all set to 95 °C pre-denaturation for 5 min, 95 °C for 45 s, 60 °C for 45 s, 72 °C for 45 s, and cycling 35 times.

### 3.6. Determination of Soil Available Nutrient Content

With reference to Jia et al. [86], the rhizosphere soils of the 45 tea tree germplasm resources were naturally air-dried indoors, ground, and sieved through a 60-mesh sieve to determine the available nitrogen, available phosphorus, and available potassium content. The available nitrogen content was determined by the alkaline dissolution diffusion method. A 1 mol/L NaOH solution was used to leach available nitrogen from the soil, which was filtered, and the filtrate was titrated using hydrochloric acid; finally, the results were converted into available soil nitrogen content. The available phosphorus content was determined by the molybdenum antimony resist colorimetric method. The soil samples were leached with 0.5 mol/L NaHCO_3_ solution and filtered, and the filtrate was added to the molybdenum–antimony colorimetric agent to develop the color. Then, the absorbance was measured at 880 nm; finally, the absorbance value was converted into the available phosphorus content of the soil. The available potassium content was determined by the flame photometer method. The soil was leached with 1 mol/L neutral ammonium acetate and filtered, the filtrate was directly measured by a flame photometer, and the results were converted into available soil potassium content.

### 3.7. Determination of Tea Quality Index Content

With reference to Zhang et al. [87], the leaves of the 45 tea germplasm resources were used and the content of tea polyphenols, theanine, and caffeine was determined. The collected tea tree leaves were bioinactivated at 105 °C for 15 min and then dried at a constant temperature of 80 °C until they reached a constant weight (36 h). After drying, the leaves were ground to a fine powder and sieved through a 60-mesh sieve before being used in the assay. Specific methods for determining the tea polyphenol content were as follows. First, a 1 g sample was weighed, 5 mL of methanol solution was added, and the mixture was heated in a 70 °C water bath for 10 min, followed by centrifugation. Then, 1 mL supernatant was taken, 5 mL Folin–Ciocalteu reagent was added, and the reaction was carried out for 5 min. Next, 4 mL of 7.5% Na_2_CO_3_ solution was added, and the reaction mixture was allowed to stand for 60 min. Finally, the absorbance was measured at 765 nm and quantified using a standard gallic acid curve. Specific methods for determining the content of theanine were as follows. First, a 1 g sample was weighed, and 100 mL of boiling distilled water was added to a water bath at 100 °C for 30 min. The extract was filtered and the volume determined. The extract was then passed through a 0.45 μm filter membrane and analyzed by high-performance liquid chromatography (HPLC), while the theanine was used to produce a standard curve for quantification. The caffeine content was determined as follows. A 3 g sample was weighed and added to 450 mL of boiling distilled water and kept in a water bath at 100 °C for 45 min. After filtration, the solution was fixed to a certain volume. Then, 10 mL of filtrate was taken and 4 mL of hydrochloric acid was added at a concentration of 0.01 mol/L, and the volume was again determined at 100 mL. After standing and filtering, the absorbance was measured at 274 nm. It was quantitatively analyzed using a stand curve composed of caffeine. For the determination of the above indexes, three independent replicates were established for each sample.

### 3.8. Statistical Analysis

The IBM SPSS Statistics software (v26) and Rstudio software (v4.2.3) were used to process and statistically analyze the data. The number of soil fungi, the content of soil available nutrients, and the content of tea quality indexes were expressed as means and standard deviations (means ± SD). The Kruskal–Wallis test was used to test the significance of the differences between the three groups at *p* < 0.05. The Wilcoxon test was used to test the significance of the differences between the two groups at *p* < 0.05. Spearman’s correlation coefficient was used to evaluate the relationship between characteristic fungi, soil available nutrients, and tea quality indexes, where *p* < 0.05 was considered a significant level of correlation. After Z-score normalization of the data, K-means clustering, heatmap generation, and PLS-SEM equation construction were performed. All graphics were produced using Rstudio software (v4.2.3).

## 4. Conclusions

In this study, fungal diversity in rhizosphere soil and available nutrient soil content and tea quality were determined in 45 tea tree germplasm resources. Did the ability of tea trees to recruit and aggregate rhizosphere soil fungi affect soil nutrient content and tea quality? We found that the ecological memory of tea tree germplasm resources for rhizosphere soil fungi consistently drove the recruitment and aggregation of dominant fungal populations in tea trees, but there were differences in their recruitment and aggregation abilities. This difference in capacity determined the content of soil available nutrients in the rhizosphere of the tea tree and determined tea quality. Thus, from the point of view of rhizosphere soil fungi, we could effectively distinguish different varieties of tea trees according to their high or low quality. Key characteristic fungi affecting soil nutrient content and leaf quality indexes were *Saitozyma*, *Archaeorhizomyce*, *Chaetomium*, and *Trechispora.* The greater the recruitment and aggregation of *Saitozyma* and *Archaeorhizomyces* by the tea tree, and the greater the repulsion of *Chaetomium* and *Trechispora*, the higher the available nutrient content of the soil and the better the tea quality. Previous studies have suggested that differences in the quality of plant germplasm resources depend on genotype. Our study suggests that a good “genotype” is only the foundation and that differences in the ability of plants to recruit and aggregate characteristic fungi are some of the most important factors in determining their quality. It is possible that the qualities of the plant will also change dramatically if it is artificially regulated to break down its original memories and instead reshape them or create new ones. Our study provides new ideas for the selection and breeding of new plant varieties and new methods for the regulation of plant cultivation, providing an important reference for the future regulation of plant cultivation through microorganisms. However, soil is a complex system, and factors such as soil physicochemical indexes and bacteria may also have some influence on fungi, which should be further explored in future studies.

## Figures and Tables

**Figure 1 plants-13-02029-f001:**
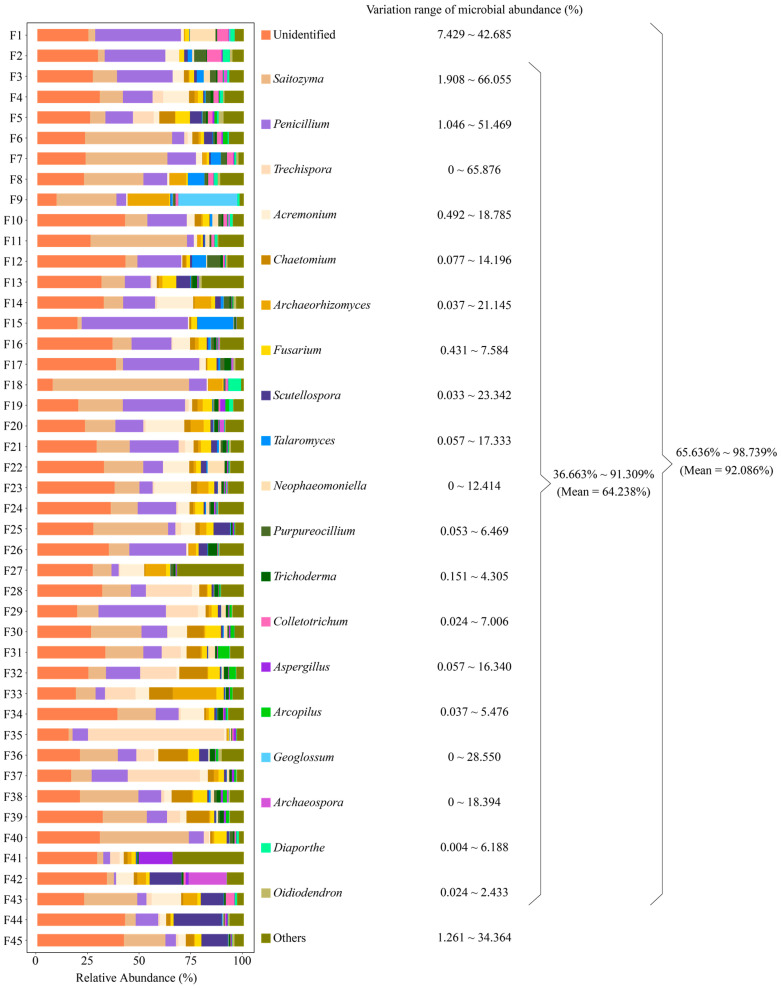
Rhizosphere soil fungal abundance analysis of 45 tea tree germplasm resources. F1~F45 denote the numbers of the 45 tea tree germplasm resources (Appendix A). Only fungal genera with relative abundance of 1% or more presented in the rhizosphere soils of the 45 tea tree germplasm resources are listed in the figure, and those that did not reach 1% or more are categorized as others. The variation range of microbial abundance indicates the range of minimum to maximum abundance of the rhizosphere soil fungi of the 45 tea tree germplasm resources.

**Figure 2 plants-13-02029-f002:**
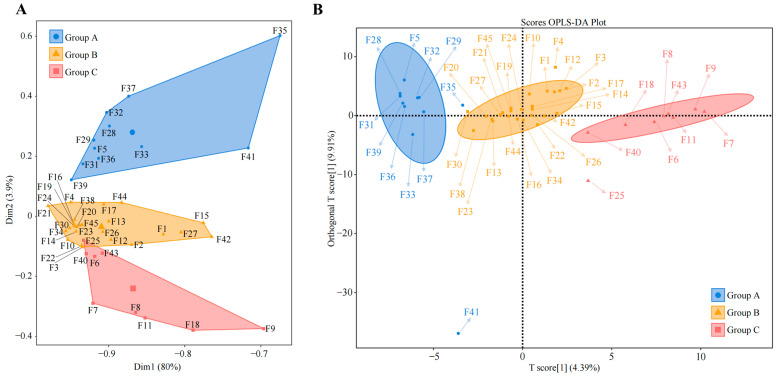
Classification and determination of tea tree germplasm resources based on the abundance of rhizosphere soil fungi. F1~F45 denote the numbers of the 45 tea tree germplasm resources (Appendix A). (**A**) Unsupervised K-means clustering was used for automatic classification based on the abundance of soil fungi in the rhizosphere of tea tree. The three categories were defined as group A, group B, and group C, and their corresponding tea tree germplasm resources are shown in Appendix A. (**B**) Simulated classification of key fungi based on rhizosphere soil fungal abundance of tea tree using the supervised OPLS-DA model (Appendix A).

**Figure 3 plants-13-02029-f003:**
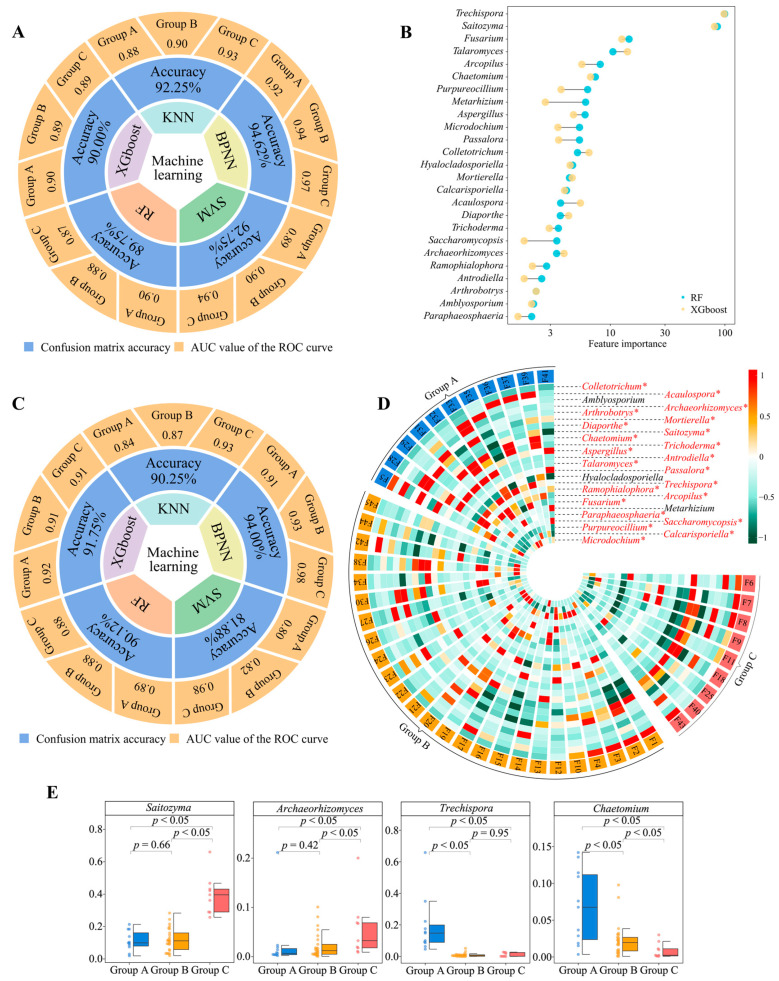
Validation of classification accuracy of key microorganisms and screening of characteristic microorganisms. F1~F45 indicate the numbers of the 45 tea tree germplasm resources (see Appendix A for the tea tree varieties corresponding to the numbers). Groups A, B, and C denote the three categories into which the 45 tea tree germplasm resources were classified after unsupervised K-means clustering (see Appendix A for the tea tree varieties corresponding to groups A, B, C). KNN, BPNN, SVM, RF, and XGboost denote five types of machine and deep learning, namely K-nearest neighbors, support vector machine, back-propagation neural network, random forest, and extreme gradient boosting. (**A**) Five machine and deep learning simulations were used to validate the accuracy of the classification based on the abundance of 181 key fungal genera. The overall accuracy of the classification was obtained through the confusion matrix, and the accuracy for groups A, B, and C was obtained through ROC curves (Appendix A). (**B**) Based on RF and XGboost, used to obtain the importance eigenvalues of different fungi when distinguishing between groups A, B, and C, 25 genera of characteristic fungi were obtained (Appendix A). (**C**) Based on the abundance of these 25 characteristic fungal genera, five machine and deep learning simulations were used to validate the accuracy of the classification. The overall accuracy of the classification was obtained through the confusion matrix, and the accuracy for groups A, B, and C was obtained through ROC curves (Appendix A). (**D**) Analysis of differences in abundance of 25 characteristic fungal genera (* indicates that the abundance of the fungus differs at the *p* < 0.05 level in groups A, B, and C; Appendix A). (**E**) Differential analysis of the four fungal genera with the highest relative abundance in groups A, B, and C.

**Figure 4 plants-13-02029-f004:**
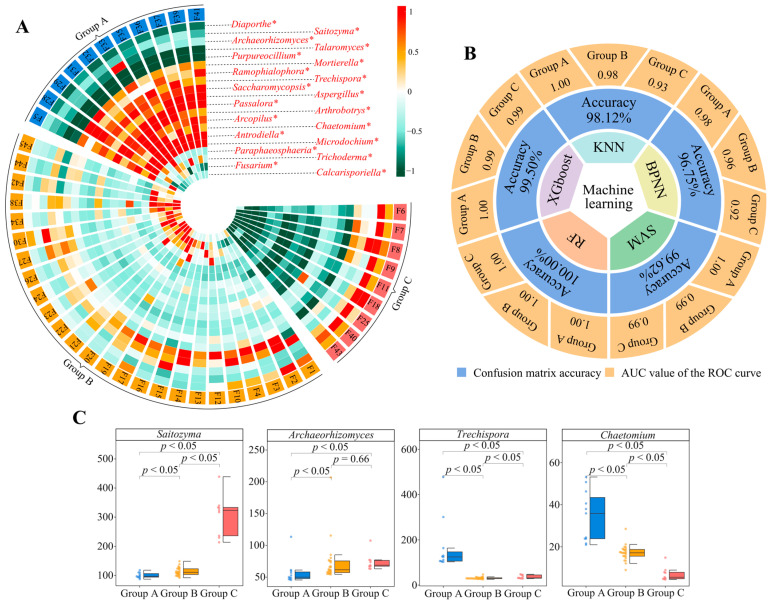
Quantitative validation of characteristic microorganisms and determination of classification accuracy. F1~F45 indicate the numbers of the 45 tea tree germplasm resources (see Appendix A for the tea tree varieties corresponding to the numbers). Groups A, B, and C denote the three categories into which the 45 tea tree germplasm resources were classified after unsupervised K-means clustering (see Appendix A for the corresponding tea varieties in groups A, B, C). KNN, BPNN, SVM, RF, and XGboost denote five types of machine and deep learning, namely K-nearest neighbors, support vector machine, back-propagation neural network, random forest, and extreme gradient boosting. (**A**) Analysis of variance of quantitative data for 20 characteristic fungal genera (* indicates that the number of these fungi that differed at the *p* < 0.05 level in groups A, B, and C; Appendix A). (**B**) Based on the quantitative data of the 20 characteristic fungal genera, five machine and deep learning simulations were used to verify the accuracy of the classification. The overall accuracy of the classification was obtained through the confusion matrix, and the accuracy for groups A, B, and C was obtained through ROC curves (Appendix A). (**C**) Differential analysis of the four fungal genera with the greatest numbers in groups A, B, and C.

**Figure 5 plants-13-02029-f005:**
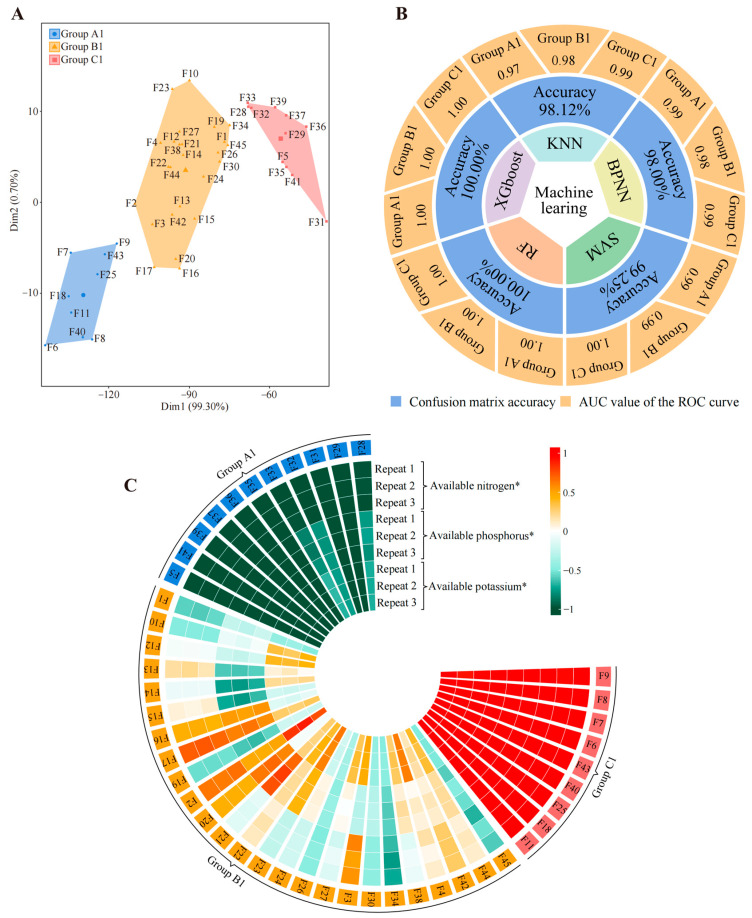
Classification of tea tree germplasm resources based on available nitrogen, phosphorus, and potassium content of soil. F1~F45 indicate the numbers of the 45 tea tree germplasm resources (see Appendix A for the tea tree varieties corresponding to the numbers). KNN, BPNN, SVM, RF, and XGboost denote five types of machine and deep learning, namely K-nearest neighbors, support vector machine, back-propagation neural network, random forest, and extreme gradient boosting. (**A**) Based on the available nitrogen, available phosphorus, and available potassium content of the tea tree rhizosphere soils (Appendix A), unsupervised K-means clustering was used for automatic classification. The three categories were defined as groups A1, B1, and C1, and their corresponding tea tree germplasm resources are shown in Appendix A. (**B**) Five machine and deep learning methods were used to simulate and verify the accuracy of the classification based on the available nitrogen, available phosphorus, and available potassium content of the tea tree rhizosphere soil (Appendix A). The overall accuracy of the classification was obtained through the confusion matrix, and the accuracy for groups A1, B1, and C1 was obtained through ROC curves (Appendix A). (**C**) Analysis of available nitrogen, available phosphorus, and available potassium content in groups A1, B1, and C1 (* indicates that the difference in the content of this index in groups A1, B1, and C1 reaches the *p* < 0.05 level; Appendix A).

**Figure 6 plants-13-02029-f006:**
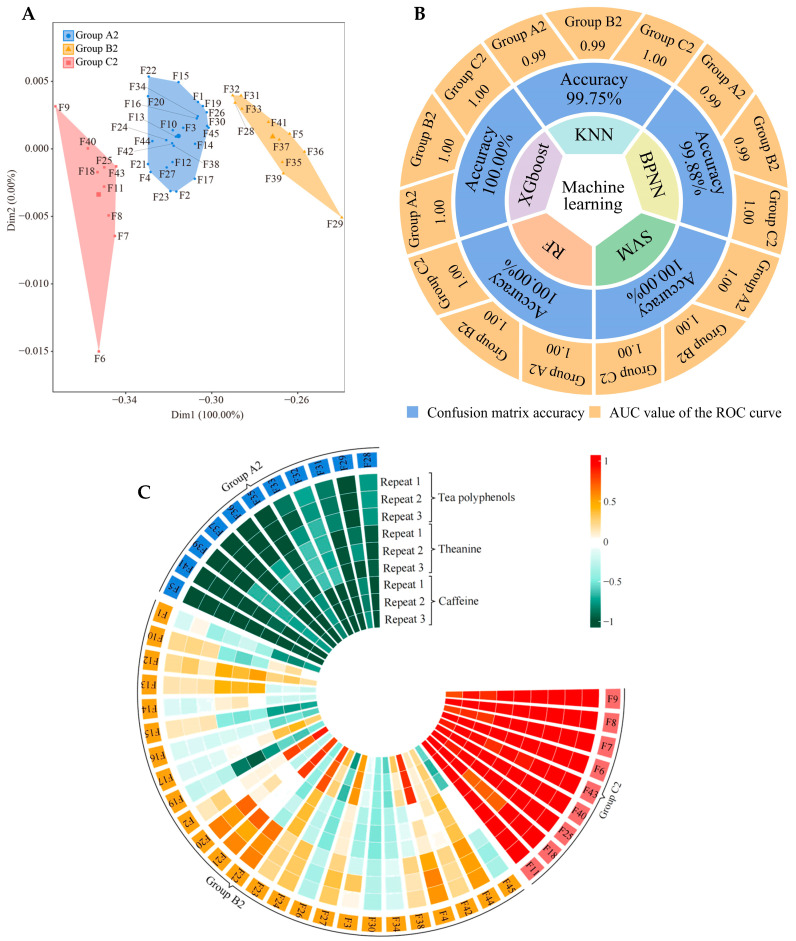
Classification of tea germplasm resources based on tea polyphenol, theanine, and caffeine content. F1~F45 indicate the numbers of the 45 tea tree germplasm resources (see Appendix A for the tea tree varieties corresponding to the numbers). KNN, BPNN, SVM, RF, and XGboost denote five types of machine and deep learning, namely K-nearest neighbors, support vector machine, back-propagation neural network, random forest, and extreme gradient boosting. (**A**) Based on the tea polyphenol, theanine, and caffeine content of the tea leaves (Appendix A), unsupervised K-means clustering was used for automatic classification. The three categories were defined as groups A2, B2, and C2, and their corresponding tea germplasm resources are shown in Appendix A. (**B**) Based on the tea polyphenol, theanine, and caffeine content of the tea leaves (Appendix A), five machine and deep learning methods were used to simulate and verify the accuracy of the classification. The overall accuracy of the classification was obtained through the confusion matrix, and the accuracy of groups A1, B1, and C1 was obtained through ROC curves (Appendix A). (**C**) Analysis of tea polyphenols, theanine, and caffeine content in groups A2, B2, and C2 (Appendix A).

**Figure 7 plants-13-02029-f007:**
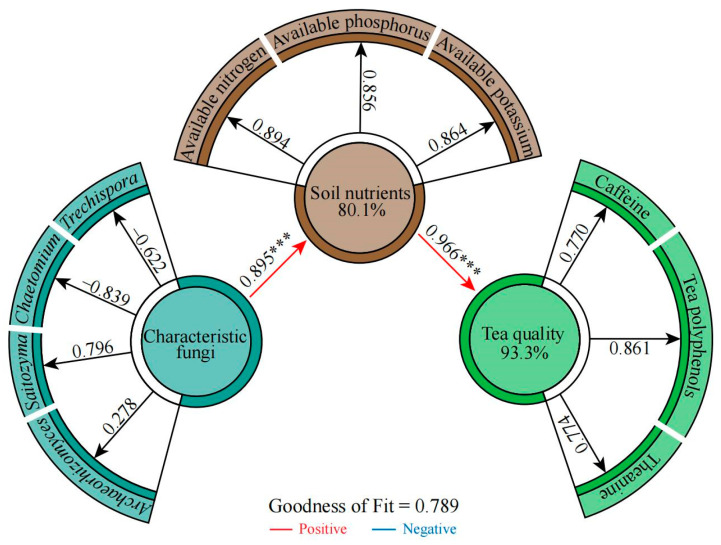
PLS-SME equations were constructed based on the quantitative data of four characteristic fungi genera; the content of soil available nitrogen, available phosphorus, and available potassium; and the content of tea polyphenols, theanine, and caffeine in tea leaves from the rhizosphere soil of 45 tea germplasm resources. Note: *** indicates that the effect value reaches the *p* < 0.001 level.

**Figure 8 plants-13-02029-f008:**
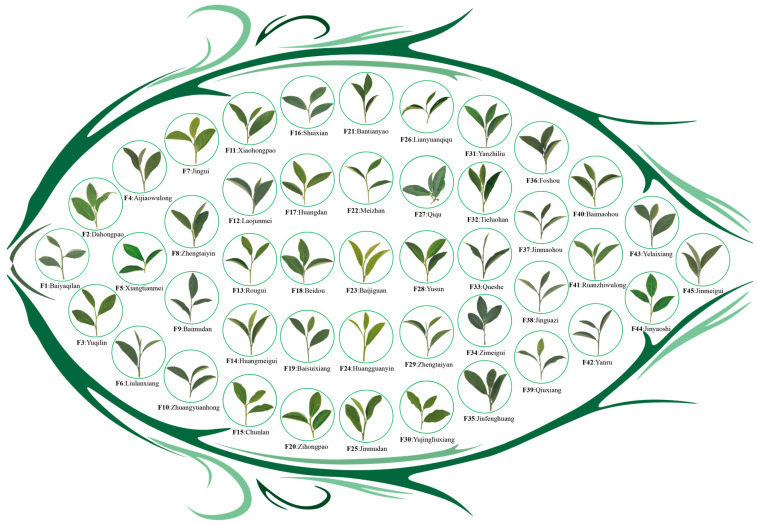
The 45 different varieties of tea germplasm resources.

## Data Availability

The ITS gene raw reads for the fungal community analyses have been deposited in the National Center for Biotechnology Information (NCBI) Sequence Read Archive (SRA) under the accession code PRJNA1045082 (https://www.ncbi.nlm.nih.gov/sra/PRJNA1045082, accessed on 26 November 2023). The data generated in this study are provided in the Appendix A. Source data and Supporting Information files are provided with this paper.

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
