# Peer review of "The Ability of Different Tea Tree Germplasm Resources in South China to Aggregate Rhizosphere Soil Characteristic Fungi Affects Tea Quality"

_plants, 2024, doi:10.3390/plants13152029_

Round 1

Reviewer 1 Report

Comments and Suggestions for Authors

Dear authors,

Please find comments and suggestions in main text pdf file 

Comments on the Quality of English Language

Author Response

A point to point response to reviewers' comments

Thank you for your professional suggestions and critical comments on our manuscript. After seriously considering the comments and suggestions of the reviewers, we have revised the manuscript.  The point-by-point responses to the suggestions and comments of reviewer are addressed below. (All changes in the article are highlighted in red by the author)

PRELIMINARY COMMENTS

  1. Please add study place such as China south or north

A: Thank you to the reviewer. The authors have added.

  1. you already mentioned in your title so please select some other keywords much more informative and positive than those mentioned in the manuscript

A: Thank you to the reviewer. The authors have modified individual keywords.

  1. you provide too long introduction

A: Thank you to the reviewer. The authors have made the introduction appropriately long in order to detail the purpose and significance of the study. Thank you very much.

  1. many scholars from where or where was research conducted in China EU or USA please provide kind of research done in some countries and why is important in China that study

A: Thank you to the reviewer. Tea tree is an important economic plant in China, which is cultivated in large areas in more than ten provinces in China, and the research on the germplasm resources of tea tree is more important to China. The authors have revised the description of the corresponding text.

  1. is possible provide some taxonomic composition of fungal diversity here? and how many family, genera and species belonging to which order and kingdom

A: Thank you to the reviewer. Because the experimental material contained 45 tea tree species, each tea tree species had some differences at different levels of rhizosphere fungal communities. In this manuscript, the authors counted the dominant fungi based on genus. And all the raw data at different levels have been uploaded to the open system and can be downloaded and queried at any time. Many thanks.

  1. are there any species name of this genus? or Please provide Gossypium sp.

A: Thank you to the reviewers. The authors have revised the manuscript to Gossypium sp.

  1. any reference?

A: Thank you to the reviewers. This is not a citation of a previous study, but the results in this study. To avoid ambiguity, the authors have deleted the expression.

  1. you dont need posed question just explain your obtained results and how was your results and what is said in other literature and show significant difference

A: Thank you to the reviewers. The authors have deleted the expression.

  1. are there name of variates of tee? it would be nice if you provide name and add one table name of cultivar tee and collection place and sample from rhizosphere soil and leaf samples and sequences information

A: Thank you to the reviewers. The authors have placed all species names in Table S1 of the Supplementary Materials. In Materials and Methods, the authors mentioned that the specific sample collection site was the Tea Tree Germplasm Resource Nursery of Wuyi University, Wuyishan City, Fujian Province, China (117°59′51.88″ E, 27°44′16.17″ N). All of the sequence information from the soil samples has been fully uploaded to the public system and is available for open download ((https://www.ncbi.nlm.nih.gov/sra/PRJNA1045082).

  1. but it is not informative fig please provide table

A: Thank you to the reviewers. The authors have placed all species names in Table S1 of the Supplementary Materials.

Reviewer 2 Report

Comments and Suggestions for Authors

The Authors 

studied the mycobiota associated to the rhizosphere of 45 diverse tea (Camellia sinensis) accessions using a metagenomic approach. They found that tea accessions clustered into three major groups based on the structure of their mycobiota. Moreover they found a strict correlation between the structure of rhizosphere mycobiota and the soil content in  nutritional elements (N, P and K) as well as between the structure o rhizosphere mycobiota and the quality of tea, which in turn  depends on  the contents  of certain chemicals in the leaves.  This part of the stusy is clear.

By contrast I have two major criticisms concerning this study:

- The first one is the description of sampling protocol. It is not clear how many plants of each tea accessions were collected. The Authors do not provide any information on the type of soil and the distribution of different tea accessions in the experimental field: were the accessions distributed randomly in the same plot or were they planted in distinct plots. The precise description of the experimental design is a very critical point.

- The second but not less important critical aspect concerns the premise and the interpretation of results. Throughout the text the Authors  stress the concept that the major driver of the quality of tea was not the genetic background but the ecological memory. However they did not report any information on the genetic background of the 45 tea accessions they examined. Moreover what the Author mean for ecological memory? According with  the literature  ecological memory is defined as the capability of the past states experienced by the organism or a community to influence the present or future ecological responses of the organism or the community, i.e. the influence of antecedent conditions on current ecological dynamics (see e.g.  Padisak 1992 and Peterson 2002). I did not find any relationship between these definitions of ecological memory  and both the experimental design and the findings of this study.

For other minor criticisms and detailed comments see notes in the text (attached PDF file)

Comments on the Quality of English Language

Author Response

A point to point response to reviewers' comments

Thank you for your professional suggestions and critical comments on our manuscript. After seriously considering the comments and suggestions of the reviewers, we have revised the manuscript.  The point-by-point responses to the suggestions and comments of reviewer are addressed below. (All changes in the article are highlighted in red by the author)

Comments and Suggestions for Authors

The Authors 

studied the mycobiota associated to the rhizosphere of 45 diverse tea (Camellia sinensis) accessions using a metagenomic approach. They found that tea accessions clustered into three major groups based on the structure of their mycobiota. Moreover they found a strict correlation between the structure of rhizosphere mycobiota and the soil content in  nutritional elements (N, P and K) as well as between the structure o rhizosphere mycobiota and the quality of tea, which in turn  depends on  the contents  of certain chemicals in the leaves.  This part of the stusy is clear.

By contrast I have two major criticisms concerning this study:

- The first one is the description of sampling protocol. It is not clear how many plants of each tea accessions were collected. The Authors do not provide any information on the type of soil and the distribution of different tea accessions in the experimental field: were the accessions distributed randomly in the same plot or were they planted in distinct plots. The precise description of the experimental design is a very critical point.

A: Thank you to the reviewer. The authors have described it in the manuscript. Three replicates were set up for experimental samples of each tea tree variety. The sampling method for each replicate was that eight plants of each tea tree variety were randomly selected and the rhizosphere soil was collected and mixed for one replicate. The leaves were collected in the same way as described above. The tea tree germplasm resources in the experiment were planted in the same area, i.e. Wuyi University Tea Tree Germplasm Resource Nursery, and planting soil was acidic red loam.

- The second but not less important critical aspect concerns the premise and the interpretation of results. Throughout the text the Authors  stress the concept that the major driver of the quality of tea was not the genetic background but the ecological memory. However they did not report any information on the genetic background of the 45 tea accessions they examined. Moreover what the Author mean for ecological memory? According with  the literature  ecological memory is defined as the capability of the past states experienced by the organism or a community to influence the present or future ecological responses of the organism or the community, i.e. the influence of antecedent conditions on current ecological dynamics (see e.g.  Padisak 1992 and Peterson 2002). I did not find any relationship between these definitions of ecological memory  and both the experimental design and the findings of this study.

A: Thanks to the reviewers. These 45 tea tree varieties are different and they have some differences in their genetic background. The authors believe that the genetic background is fundamental, and of course, the concept of ecological memory, as mentioned by Padisak 1992 and Peterson 2002, is strongly endorsed by the authors, and the authors concur with the reviewers' understanding of this concept. The above is not in conflict with this study, which is concerned with the ecological memory between tea trees and rhizosphere soil fungi, and in particular the difference in ecological memory across 45 different tea tree germplasm resources. Regarding the relationship between the results and ecological memory, the authors have revised the manuscript and adjusted the presentation according to the expert's comments. We thank the experts for their careful review of the manuscript again.

For other minor criticisms and detailed comments see notes in the text (attached PDF file)

  1. REDUNDANCY . THIS SENTENCE CAN BE DELETED. AVOID COLLOQUIAL SENTENCES IN A SCIENTIFIC PAPER

A: Thanks to the reviewers. The authors have removed it.

  1. However also ecological

A: Thanks to the reviewers. The authors have revised it.

  1. DIFFERED FROM WHAT? AMONG EACH OTHER? REPHRASE

A: Thanks to the reviewers. The authors have revised it.

  1. Artificial inoculation of rhzosphere fungi may rephase ecological memory of tea plants improving their quality substantially.

A: Thanks to the reviewers. The authors have revised it.

  1. Camellia sinensis

A: Thanks to the reviewers. The authors have revised it.

  1. 1st DEFINITION, 2nd DEFINITION

A: Thanks to the reviewers. The authors have revised it.

  1. Plants

A: Thanks to the reviewers. The authors have revised it.

  1. and

A: Thanks to the reviewers. The authors have revised it.

  1. system

A: Thanks to the reviewers. The authors have revised it.

  1. interacting with the plant

A: Thanks to the reviewers. The authors have revised it.

  1. DELETE IT IS ALREADY IN THE SPECIFIC NAME

A: Thanks to the reviewers. The authors have revised it.

  1. PLEASE REPHRASE TO CLARIFY WHAT YOU MEAN.

A: Thanks to the reviewers. The expression was somewhat colloquial and the authors removed it.

  1. HAS NOT A GOOD GENETIC BACKGROUND

A: Thanks to the reviewers. The authors have revised it.

  1. IT IS WELL KNOWN THAT THE PERFORMANCE OF CROPS IS THE RESULT OF THE INTERACTION BETWEEN THE GENOTYPE AND THE ENVIRONMENT.  THIS DOES NOT MEAN NECESSARY THE PLANTS HAVE A MEMORY.  PLEASE, EXPLAIN BETTER THIS CONCEPT.

A: Thanks to the reviewers. The authors have rephrased and highlighted the place in red.

  1. By contrast

A: Thanks to the reviewers. The authors have revised it.

  1. rhizosphere soil and leaf samples were collected from

A: Thanks to the reviewers. The authors have revised it.

  1. I DO NOT THINK THAT THE RHIZOSPHERE EFFECT CAN ALWAYS REFERRED TO THE CONCEPT OF ECOLOGICAL MEMORY

A: Thanks to the reviewers. The authors have made changes to the expression here. We hope to receive the expert's approval.

  1. AGAIN I DO NOT UNDERSTAND HOW YOU DEMOSTRATED WITH YOUR FINDINGS THAT BOTH PLANTS AND FUNGI POSSES ECOLOGICAL MEMORY.

A: Thanks to the reviewers. The authors have made changes to the expression here. We hope to receive the expert's approval.

  1. THERE IS SOMETHING MISSING IN THIS SENTENCE

A: Thanks to the reviewers. The authors have made changes to the expression here. We hope to receive the expert's approval.

  1. I WOULD SAY ALSO IN THE TYPE AND RELATIVE ABUNDANCE OF........

A: Thanks to the reviewers, the authors have revised the expression and added the corresponding fungi.

  1. these fungi

A: Thanks to the reviewers. The authors have revised it.

  1. overlapped

A: Thanks to the reviewers. The authors have revised it.

  1. REPHRASE

A: Thanks to the reviewers. The authors have made changes to the expression here. We hope to receive the expert's approval.

  1. PER EACH TEA VARIETY?

A: Thank you to the reviewer.Yes, the authors added the description.

  1. THREE REPLICATE PER TREE.

DOES IT MEAN 3 REPLICATES X 8 TREES PER EACH TEA VARIETY (TOTAL 24 REPLICATES PER TEA VARIETY) ?

A: Thank you to the reviewer. The authors described 8 randomly selected plants of each tea tree variety and collected rhizosphere soil mixed as one replicate. Sampling of tea tree leaves was also done by mixing the leaves of 8 tea trees for 1 replicate.

  1. in rhizosphere soil

A: Thanks to the reviewers. The authors have revised it.

  1. I STILL DO NOT UNDERSTAND THE CONNECTION WITH ECOLOGICAL MEMORY THE AUTHORS REFER TO THROUGHOUT THE ENTIRE TEXT

A: Thanks to the reviewers. The authors have made changes to the expression here. We hope to receive the expert's approval.

  1. MAY BE THE AUTHORS WANT TO SAY THERE IS A CORRELATION BETWEEN THE RHIZOSPHERE MYCOBIOTA OF TEA PLANTS AND THE QUALITY OF TEA.

THIS CORRELATION WAS POSITIVE FOR Saitozyma AND  Archaeorhizomyces

WHILE IT WAS INVERSE FOR Chaetomium  AND Trechispora GENERA.

  WHILE IT WAS INVERSE FOR

A: Thank you to the reviewer. Yes.The authors first expressed that the ability of the tea tree to aggregate these fungi affected its own quality. Then they expressed that which fungi were added and which fungi were reduced could affect the quality of the tea tree.

  1. depend on genotype

A: Thanks to the reviewers. The authors have revised it.

  1. SEE PREVIOUS COMMENTS ON ECOLOGICAL MEMORY

A: Thanks to the reviewers. The authors have made changes to the expression here. We hope to receive the expert's approval.

Reviewer 3 Report

Comments and Suggestions for Authors

The manuscript presents research results on the relationship between the ability of various tea tree genotypes to aggregate rhizosphere fungi and the quality of tea. The topic is very interesting because it concerns the ecological memory of plants and rhizosphere soil fungi and allows for obtaining new knowledge. This issue is also very important because it shows links between the colonization of the rhizosphere by various genera of fungi and the quality of tea, which is very important for practice.

The great advantage of the presented research results is the use of the latest techniques used in metagenomics for analysis, such as high-throughput sequencing and deep machine learning, which enable experiments to be carried out on large populations of microorganisms, analysis of very large amounts of data and demonstration of connections between individual components of the analysis.

The authors of the manuscript used fungal ITS amplicon sequencing using Illumina platform for research, and carried out construction and evaluation of deep machine learning models. In order to quantify 20 types of fungi, the qRT-PCR technique was used, the content of nutrients in the soil and the content of the tea quality index were determined.

The most important result obtained by the authors of the manuscript is the demonstration of the relationship between the ability of tea tree to aggregate fungi from the genera Saitozyma, Archaeorhizomyces, Chaetomium and Trechispora and the content of available nutrients in the soil and the quality of tea. Based on the results, the authors concluded about the ecological memory between tea trees and soil fungi, which leads to differences in their ability to recruit and aggregate different genera of fungi.

The comments concern the spelling of the names of the Shannon-Wiener and Simpson coefficients and the Folin-Ciocialteu reagent, which should be corrected. Additionally, the manuscript contains typographical errors of words in lines 242 and 392.

The manuscript contains new, valuable results, and publications cited in the text are placed in References. The manuscript is clearly written and should be published.

Author Response

A point to point response to reviewers' comments

Thank you for your professional suggestions and critical comments on our manuscript. After seriously considering the comments and suggestions of the reviewers, we have revised the manuscript.  The point-by-point responses to the suggestions and comments of reviewer are addressed below. (All changes in the article are highlighted in red by the author)

Comments and Suggestions for Authors

The manuscript presents research results on the relationship between the ability of various tea tree genotypes to aggregate rhizosphere fungi and the quality of tea. The topic is very interesting because it concerns the ecological memory of plants and rhizosphere soil fungi and allows for obtaining new knowledge. This issue is also very important because it shows links between the colonization of the rhizosphere by various genera of fungi and the quality of tea, which is very important for practice.

A: We thank the reviewers for recognizing the research results of the author team.

The great advantage of the presented research results is the use of the latest techniques used in metagenomics for analysis, such as high-throughput sequencing and deep machine learning, which enable experiments to be carried out on large populations of microorganisms, analysis of very large amounts of data and demonstration of connections between individual components of the analysis.

A: We thank the reviewers for recognizing the research results of the author team.

The authors of the manuscript used fungal ITS amplicon sequencing using Illumina platform for research, and carried out construction and evaluation of deep machine learning models. In order to quantify 20 types of fungi, the qRT-PCR technique was used, the content of nutrients in the soil and the content of the tea quality index were determined.

The most important result obtained by the authors of the manuscript is the demonstration of the relationship between the ability of tea tree to aggregate fungi from the genera Saitozyma, Archaeorhizomyces, Chaetomium and Trechispora and the content of available nutrients in the soil and the quality of tea. Based on the results, the authors concluded about the ecological memory between tea trees and soil fungi, which leads to differences in their ability to recruit and aggregate different genera of fungi.

The comments concern the spelling of the names of the Shannon-Wiener and Simpson coefficients and the Folin-Ciocialteu reagent, which should be corrected. Additionally, the manuscript contains typographical errors of words in lines 242 and 392.

A: Thank you to the reviewers. The authors have revised and highlighted in red.

The manuscript contains new, valuable results, and publications cited in the text are placed in References. The manuscript is clearly written and should be published.

A: We thank the reviewers for recognizing the research of the author's team. The authors have carefully revised and highlighted in red all the relevant information that the reviewers have annotated in the manuscript. Thank you again to the reviewers.